# Active Learning Neural C-space Signed Distance Fields for Reduced Deformable Self-Collision

Xinhao Cai[*]
McGill University

Eulalie Coevoet[†]
McGill University

Alec Jacobson[‡]
University of Toronto

Paul Kry[§]
McGill University

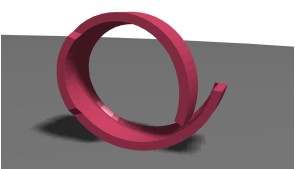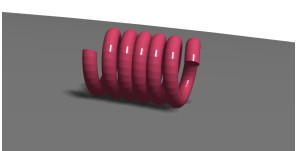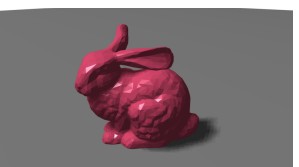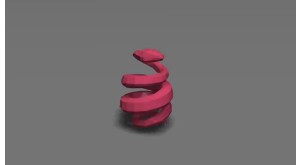

Figure 1: Examples from our supplementary video, showing self collision for the bracelet, spring, bunny, and snake models. Self-collision is identified using a learned neural network SDF, and collision response uses the SDF gradient computed via back-propogation within a constraint solver. Neural SDFs work well for low dimension reduced spaces (e.g., the bracelet and spring with dimension 3), while models that need more dimensions to provide good reduced deformation models (e.g., the bunny with dimension 10, and the snake with dimension 7) have much less accurate learned collision manifolds.

## ABSTRACT

We present a novel method to preprocess a reduced model, training a neural network to approximate the reduced model signed distance field using active learning technique. The trained neural network is used to evaluate the self-collision state as well as the self-collision handling during real time simulation. Our offline learning approach consists of two passes of learning. The first pass learning generates positive and negative point cloud which is used in the second pass learning to learn the signed distance field of reduced subspace. Unlike common fully supervised learning approaches, we make use of semi-supervised active learning technique in generating more informative samples for training, improving the convergence speed. We also propose methods to use the learned SDF function in real time self-collision detection and assemble it in the constraint Jacobian matrix to solve the self-collision.

**Index Terms:** reduced model—self-collision—configuration space—signed distance field; active learning—contact constraint

## 1 INTRODUCTION

In computer animation, simulating the physics of models usually requires solving large linear systems whose size conforms to the generalized coordinate of the model, and this can be costly if the model consists of a huge number of vertices. Model reduction [20] is a technique used to approximate the simulation of the full dynamic system with a simplified one by projecting the high-dimensional system onto low-dimensional subspace. With much fewer variables in the reduced system, the equations can be solved much quicker while maintaining high fidelity of the original system. Reduced model deformation is an application of model reduction in computer animation to improve efficiency and is very relevant to applications such as games and training simulations, where a real-time computation is required.

Reduced model deformation can simplify the dynamic system solving, but the complexity of self-collision detection of the reduced

---

[*]e-mail: xinhao.cai@mail.mcgill.ca

[†]e-mail: eulalie.coevoet@gmail.com

[‡]e-mail: jacobson@cs.toronto.edu

[§]e-mail: kry@cs.mcgill.ca

model is still related to the complexity of the model mesh, since we must test each pair of the triangle mesh. Although a number of algorithms or data structures have been proposed to speed up the self-collision detection by culling unnecessary tests, like the BD-Tree [13] and various culling strategies, some triangle-triangle intersection tests are still inevitable.

In this paper, we assume a reduced deformation model and focus on efficient evaluations of self-collision. We learn a function to approximate the configuration space (C-space) signed distance field of a reduced model, which is a mapping from the model's configuration to the signed distance to the model's closest self-collision state. The idea is inspired by the ellipsoid bound used by Barbic and James [5] to conservatively rule out self collision, but extends that implicit function to a more complex function that represents the actual collision boundary rather than a recursive application of conservative bounds as computed by Barbic and James. We show that in many cases a single, inexpensive function can replace the collision hierarchy, while also providing the gradients necessary to compute a collision response.

Using traditional supervised learning methods in this case poses two challenges. First, the actual C-space self-collision boundary is unknown. Given a random deformation configuration, the only information we can easily compute is the sign (i.e., whether the model is in self-collision or not), so there are no ground truth signed distance values to be used in training. Second, as more modes are used to deform a model, the number of dimensions of the C-space increases, and with naive uniform sampling the number of samples needed to learn the C-space boundary increases exponentially. In order to overcome these difficulties, we use approximated signed distance and eikonal loss terms to help the neural network function learn the C-space signed distance field. We also use active learning as our learning strategy for efficient sampling.

Active learning is a type of semi-supervised learning where the learner automatically chooses the most informative data to label for training, which can improve the convergence of training. With active learning, the selected training data tends to distribute around the ground truth self-collision boundary, so we harvest the point cloud based on this observation and use that to approximate the signed distance value of a given configuration.

The contribution of this paper is to explore a new way to pre-process reduced deformable models, using active learning to learn the self-collision signed distance field (SDF) in C-space. We also show how to use the learned SDF function in real-time self-collision

detection and self-intersection handling during physics simulations.

## 2 RELATED WORK

Our work is based on reduced deformable models. We learn reduced C-space SDF of reduced models, and use the trained neural network in self-collision detection and self-collision handling. The initial model reduction applications [9, 12, 18, 20] in computer animation are based on linear systems. Since the linear elastic internal forces are computed using the rest shape stiffness matrix, the deformation produces noticeable artifacts when the model has large deformation. In order to relieve the distortion produced by large deformation, Barbič and James [3, 4] investigate St.Venant-Kirchhoff deformable models with elastic forces which are cubic polynomials in reduced coordinate and provide methods to evaluate the elastic forces in real time. In addition to solid deformable models, model reduction is also used in acoustic simulations [6, 11] and fluid simulations [22, 24].

Self-collision detection (SCD) has been widely studied in computer animation. Bounding volume hierarchies (BVH) is the most commonly used data structure both in inter-object collision detection and SCD [21]. For cloth surfaces, Volino and Thalmann [23] use an improved hierarchical representation, taking advantage of geometrical regularity to skip SCD between large surface regions that are close, yet impossible to contact. The approaches for improving the speed of SCD have mainly focused on two techniques: improving BVH updates, and culling unnecessary BV node tests. For improving BVH updates, Larsson and Akenine-Möller [15] propose a hybrid update method using a combination of an incremental bottom-up update and a selective top-down update. Then, they blend associated sets of reference bounding volumes to enable lazy BVH updates [16]. James and Pai [13] propose the bounded deformation tree (BD-Tree), which makes use of the information on deformation modes, and updates the bounding sphere conservatively. For culling unnecessary tests, subspace self-collision culling [5] is proposed, in which a conservative certificate in C-space is precomputed and used to rule out some tests. Energy based self-collision culling certificates [26] have also been proposed by exploiting the idea that a mesh cannot self collide unless it deforms enough.

Machine learning (ML) methods build models based on sample data. A trained model can serve as a fast approximation of the studied problem. Practically, machine learning has been used abundantly in the fields of robotics, geometry processing, and computer animation and works well as a black box algorithm. Jiang and Liu [14] use a fully connected neural network to fit human motion with limits such as self contact, and they use network gradients to define constraint directions, which is an inspiration to our self-collision response. Neural networks are also used in geometry reconstruction. Atzmon and Lipman [1] propose a sign agnostic learning (SAL) method, in which an unsigned loss function was used to learn the signed distance field defined by the geometry. SAL is later improved into SALD [2], where derivatives of the loss term are incorporated into regression loss, which is the inspiration of our eikonal term in the loss function. Similar to our work on self collision, Zhesch et al. [25] also propose neural collision detection for reduced deformable models with a focus on collision between objects.

Machine learning has also been used for learning the physics of animation. Fulton et al. [8] use autoencoder neural networks to produce reduced model dynamics. Holden et al. [10] propose a data-driven reduced model physics simulation method, which includes the collision response, and satisfies memory and performance constraints imposed by modern interactive applications. One of the machine learning techniques that interests us is active learning. Active learning automatically chooses samples to label and thus can improve the convergence rate compared with regular supervised learning. Pan et al. [17] propose an active learning approach to learn the C-space boundary between rigid bodies and use the boundary to approximate global penetration depth. In our work, labeling the

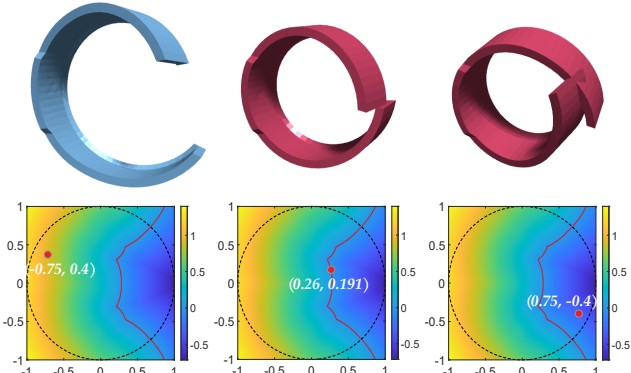

Figure 2: Bracelet model C-space with 2 modes, showing a configuration in the free space (left), a contact configuration on the boundary (middle), and a configuration involving interpenetration (right).

samples requires performing self-collision detection on the model, and this can be expensive in the time cost depending on the complexity of the model. So, active learning is a key part in our work since it largely reduces the number of samples that require labeling. In contrast to Pan et al., instead of using SVM to learn the boundary then obtaining the distances in real-time, we use neural networks to directly learn the SDF.

## 3 REDUCED MODEL C-SPACE SIGNED DISTANCE

The deformation of a reduced model is represented by a reduced coordinate (or deformation configuration) $\mathbf{q} \in \mathbb{R}^r$, where $r$ is the number of deformation modes. Then, the full coordinate of the vertices displacement is reconstructed by $\Delta \mathbf{x} = \mathbf{U}\mathbf{q}$, where each column of the matrix $\mathbf{U} \in \mathbb{R}^{3n \times r}$ is a deformation mode. The $r$-D space where the configuration $\mathbf{q}$ lives is the configuration space (C-space).

The signed distances in C-space are determined by a self-collision boundary $T_{\text{bound}}$, which is a collection of points that make a reduced model deform to just having self-contact. The boundary divides the C-space into collision space $T_{\text{collision}}$ where the configurations generate self-intersections and free space $T_{\text{free}}$ where the model is free from self-collision.

**Sign:** The sign represents if the model is self-collision free. We use $t(\mathbf{q}) \in \{1, -1\}$ to denote the target sign of the given configuration $\mathbf{q}$. If $\mathbf{q}$ makes the model in self-intersection or just touch itself then $t(\mathbf{q}) = -1$ otherwise $t(\mathbf{q}) = 1$. All the positive signs form the free space $T_{\text{free}} = \{\mathbf{q} \mid t(\mathbf{q}) = 1\}$ and all the negative signs form the collision space $T_{\text{collision}} = \{\mathbf{q} \mid t(\mathbf{q}) = -1\}$.

**Distance:** The distance is defined as the Euclidean distance from the configuration to the closest point in $T_{\text{bound}}$, i.e., $d(\mathbf{q}) = \min_{\mathbf{q}^* \in T_{\text{bound}}} ||\mathbf{q} - \mathbf{q}^*||_2$.

Figure 2 shows the plot of example configurations in the 2D C-space of a bracelet model, as well as the geometry under the deformation. When $\mathbf{q}$ causes the model to just touch itself, $\mathbf{q}$ is on the self-collision boundary (Figure 2 middle), which is highlighted in a red line. The colored signed distance field (SDF) shows the closest Euclidean distance to the boundary. The self-collision boundary and the SDF are what we want to learn with neural networks. Note that a reduced model may have more than two modes, and in that case the target collision boundary and SDF live in an $r$-D C-space where $r$ is the number of modes.

The dashed line in Figure 2 shows an equal-energy level set on which the configurations produce the same elastic energy. Intuitively, the model does not deform enough to produce self-contact unless it reaches a certain amount of elastic energy, so it would be reason-

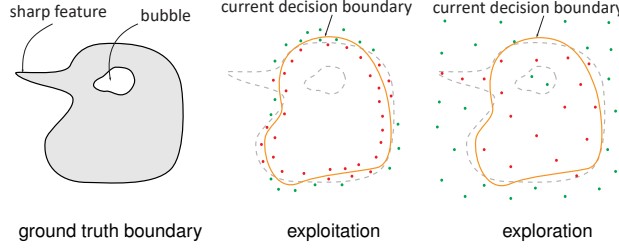

sharp feature · bubble · current decision boundary · current decision boundary

ground truth boundary · exploitation · exploration

Figure 3: Exploitation samples near the boundary help improve local accuracy, while exploration samples help identify missing parts of the boundary.

able to sample within an equal-energy bound during training. The equal-energy level set here is a sphere since the deformation modes are simply obtained from LMA, but it can become irregular if the deformation modes are obtained from modal derivatives or manual selections. For generality, we set the training and sampling domain to be a $2^r$ hyper cube with each dimension limited within $[-1, 1]$. In order to make sure the configurations during simulation are safely included by the sampling domain, we simulate the model to collect the maximum absolute value of each configuration entry and scale the deformation bases before the learning process.

## 4 ACTIVE LEARNING C-SPACE SDF

We use a two-pass active learning algorithm to train a neural network to represent a C-space SDF with the sample labels only consisting of signs. The network $f(\mathbf{q})$ takes the configuration as input, and its output is a scalar value approximating the signed distance to the closest self-collision state. In the first pass, we use active learning to learn the collision boundary, but our main goal is to cache the growing training set as a point cloud. In the second pass, we train the neural network to learn the SDF using the cached point cloud in the first pass.

### 4.1 Two-Pass Active Learning Overview

Active learning is a semi-supervised machine learning approach. It has been used by Pan et al. [17] to learn inter-object rigid body C-space and achieve great success. During the training process, an active learner continuously chooses samples from an unlabeled data pool, and the selected data are labeled to train the machine learning model.

In active learning, exploitation and exploration strategies are used to choose samples. Figure 3 shows an example of exploitation samples and exploration samples. Exploitation is good at selecting data that are close to the current decision boundary and helps efficiently refine the decision boundary, but it can also cause serious sample bias and consequently poor performance. Exploration is good at shaping the overall structure of the decision boundary and selecting samples in undetected regions, but it can also cause serious sample bias and consequently poor performance.

For learning the C-space SDF of a reduced model, we perform active learning twice, and each with different purposes. The first pass generates a point cloud where all the samples are divided according to their signs. For the second pass, this point cloud is used to compute approximated signed distances by looking for the closest points to the samples. In the following discussions, we use subscript $i$ to denote the learning iteration, superscript $(k)$ to denote a sample index inside a batch, and $f$ to represent the neural network function.

Both passes go through a fixed number of iterations to train a neural network to fit the collision boundary. For the first pass, the loss function to optimize consists of a sign loss term $L_{\text{sign}}$, and an eikonal loss term $L_{\text{eik}}$. At each iteration of the first pass, we first generate adaptive training samples $\mathbf{Q}_i$ using both exploration and

exploitation strategy, and query for their signs $T_i$ by performing SCD. Then we add the generated samples $(\mathbf{Q}_i, T_i)$ to the adaptive training batch $(\mathbf{Q}_a, T_a)$, which is maintained and grows at each iteration. The adaptive training batch corresponds to the sign loss term $L_{\text{sign}}$ that measures the sign predictions error. In addition, eikonal samples $\mathbf{Q}_{\text{eik}_i}$ are randomly generated at each iteration, which correspond to the eikonal loss term $L_{\text{eik}}$ that constrains the gradient magnitude. The neural network $f$ is then trained a user-defined number of epochs $n$ with an Adam optimizer. Note that we use incremental training, which means the neural network begins step $i$ with the trained network from step $i - 1$. After the first pass training is finished, the adaptive training batch is then divided according to the signs of the samples to be the cached point cloud.

In addition to $L_{\text{sign}}$ and $L_{\text{eik}}$, the loss function for the second pass has a signed distance loss $L_{\text{sign}}$ that trains the neural network to learn the signed distances. The adaptive training batch and the eikonal samples are generated in the same way as in the first pass. Additionally for the second pass, we uniformly generate samples $\mathbf{Q}_{\text{sd}_i}$ and query the input point cloud for their approximated signed distances $D_i$. Then the neural network is trained with the adaptive training batch $(\mathbf{Q}_a, T_a)$, eikonal samples $\mathbf{Q}_{\text{eik}_i}$, and the signed distance batch $(\mathbf{Q}_{\text{sd}_i}, D_i)$

### 4.2 Exploration Samples

Exploration samples serve as detecting regions, bubbles, and sharp features that are unrecognized by the current network. In our approach, we uniformly generate $N_{\text{explore}}$ random configurations $\mathbf{Q}_{\text{rand}_i}$, and use the current network to predict the signs. If the predicted sign is wrong, we add the sample to the training batch. So at each step, the exploration sample batch is

$$\mathbf{Q}_{\text{exploration}_i} = \left\{ \mathbf{Q}_{\text{rand}_i}^{(k)} \mid f(\mathbf{Q}_{\text{rand}_i}^{(k)}) t(\mathbf{Q}_{\text{rand}_i}^{(k)}) < 0 \right\}, \quad (1)$$
$$\text{where} \quad k \in [1, N_{\text{explore}}] \, .$$

Batch size $N_{\text{explore}}$ can be relatively small (we choose $N_{\text{explore}} = 500$ in tests) since the sign query can be expensive depending on the complexity of the model, and in practice the accumulation of the exploration samples does help in detecting bubbles and sharp features.

### 4.3 Exploitation Samples

Exploitation samples help refine the network sign decision boundary and push the prediction boundary $\{\mathbf{q} \mid f(\mathbf{q}) = 0\}$ closer to the ground truth $T_{\text{bound}}$. As the learning progresses, the exploitation samples tend to focus around $T_{\text{bound}}$. In our approach, we first uniformly generate random configuration pool $\mathbf{Q}_{\text{pool}_i}$ with $N_{\text{pool}}$ samples. Then we find candidate samples $\mathbf{Q}_{\text{cand}_i}$ that are closest to the current prediction boundary. The top $N_{\text{cand}}$ samples that have the highest scores, which are computed by

$$\text{score} = \frac{1}{1 + \left| f(\mathbf{Q}_{\text{pool}_i}^{(k)}) \right|} \, , \quad (2)$$

are picked as candidates $\mathbf{Q}_{\text{cand}_i}$. From $\mathbf{Q}_{\text{cand}_i}$, we pick the samples with wrong sign predictions and additionally $N_{\text{extra}}$ samples with the highest scores to form exploitation samples $\mathbf{Q}_{\text{exploit}}$.

In practice, $N_{\text{pool}}$ can be large (we choose $N_{\text{pool}} = 50000$) since evaluating the network output is cheap. Batch sizes $N_{\text{cand}}$ and $N_{\text{extra}}$ are relatively small, and we set $N_{\text{cand}} = 500$ and $N_{\text{extra}} = 80$ in our tests.

### 4.4 Eikonal Samples

The eikonal samples correspond to an eikonal loss term $L_{\text{eik}}$ in the loss function that imposes constraints to the gradient magnitude of the neural network function. The eikonal samples are uniformly

drawn in the domain, and are drawn at each iteration to assist the learning in producing a function with unit gradient, so that the neural network not only learns the self-collision boundary, but also the Euclidean distance to the boundary.

Since the neural network aims at learning the SDF in the whole C-space, the gradient magnitude constraint should be uniformly applied everywhere within the sample domain. However as the number of modes increases, the number of uniform eikonal samples needed to abundantly spread across the sample domain increases exponentially, and the time cost of computing the eikonal loss increases as well. Therefore, we borrow the idea from stochastic gradient descent, and randomly draw $N_{\text{eik}}$ eikonal samples at each step (we set $N_{\text{eik}} = 5000$ in our tests). At each step, the eikonal loss is computed from the new eikonal samples, such that the gradient magnitude stochastically converges to the desired range.

The eikonal loss uses the magnitude of the network gradient with respect to the input, and is used to apply a penalty when the gradient magnitude is not 1 or within the range set by the user. The eikonal term is computed by

$$L_{\text{eik}} = \frac{1}{N_{\text{eik}}} \sum_{k=1}^{N_{\text{eik}}} h\left(|\nabla f(\mathbf{Q}_{\text{eik}_i})|\right) , \tag{3}$$

$$\text{where} \quad h(x) = \begin{cases} x + \frac{1}{x} & 0 < x < 1, \\ 2 & 1 \leq x \leq 1 + \xi, \\ x - \xi + \frac{1}{x - \xi} & x > 1 + \xi. \end{cases} \tag{4}$$

In the eikonal loss, we use a piecewise loss function $h(x)$ that causes an infinitely large penalty when the gradient magnitude is 0 or $+\infty$, and less loss penalty when the gradient magnitude approaches a biased region near 1. In practice, we set $\xi = 0.2$. The biased region is set slightly larger than 1 because we want the trained neural network to be more decisive around the decision boundary, which means the trained SDF around the decision boundary should have a larger gradient magnitude rather than a smaller one. This is because when the trained neural network is used in self-collision response, the queried configuration is always off the boundary due to time discretization, and we want to make sure the gradient is always (at least generally) pointing towards the closest point on the boundary. If the trained SDF generally has small gradient size, it is likely the gradient direction slightly off the boundary is messed up and pointing to a random direction.

One of the challenges in the implementation of the eikonal loss is that second order derivatives of the NN function are needed to optimize the loss. According to the chain rule, the gradient used to update the weights within the nerual network includes two parts. First we need to compute the derivative of the eikonal loss function. Then we need to compute the gradient of the eikonal loss argument $\nabla f(\mathbf{Q}_{\text{eik}_i})$ with respect to the network weights, which is a second order derivative. Some of the neural network tools do not support back propagation for computing second order derivatives. In our implementation, the network gradient $\nabla f(\mathbf{Q}_{\text{eik}_i})$ is computed using finite differences, so that the second order derivatives can be treated like first order derivatives and computed by back propagation.

### 4.5 First Pass Learning (Point-cloud Generation Pass)

The first pass treats the input training data as a binary classification problem, but the main purpose is to collect the point cloud, which is used to generate approximated signed distance data in the second pass.

**Point Cloud Collection:** The point cloud is meant to be used to generate approximated signed distance, so it's samples need to densely spread around the whole self-collision boundary, not missing any bubbles or sharp features. The adaptive training samples picked by exploitation and exploration naturally meet the demand, so we use the adaptive training batch in the first pass as the point cloud. We divide the adaptive training batch $(\mathbf{Q}_a, T_a)$ into positive cloud and negative cloud according to the signs of the samples, then $\mathbf{Q}_{\text{pos}}$ and $\mathbf{Q}_{\text{neg}}$ are

$$\mathbf{Q}_{\text{pos}} = \left\{ \mathbf{Q}_a^{(k)} \mid T_a^{(k)} = 1, k \in [1, N_a] \right\} , \tag{5}$$

$$\mathbf{Q}_{\text{neg}} = \left\{ \mathbf{Q}_a^{(k)} \mid T_a^{(k)} = -1, k \in [1, N_a] \right\} , \tag{6}$$

where $N_a$ is the number of samples in the adaptive training batch.

**First Pass Loss Function:** The loss $L$ of the first pass consists of sign loss and eikonal loss, which is computed by

$$L = L_{\text{sign}} + \lambda L_{\text{eik}} . \tag{7}$$

The eikonal loss term is evaluated according to the gradient magnitude of the network, which is discussed in Section 4.4.

The sign loss penalizes the unmatching signs between predicted signs and target ones. It compares the signs between the prediction and the ground truth, and samples only contribute to this loss term when the signs do not match. The sign loss is computed by

$$L_{\text{sign}} = \sum_{k=1}^{N_a} S(|f(\mathbf{Q}_a)|)_k \max \left\{ -T_a^{(k)} (2\sigma(f(\mathbf{Q}_a^{(k)})) - 1), 0 \right\}, \tag{8}$$

$$\text{where} \quad S(x)_i = \frac{e^{-x^{(i)}}}{\sum_{j=1}^{N_a} e^{-x^{(j)}}} . \tag{9}$$

Here we apply a softmax weighting function $S(x)_i$ to have more importance on the samples closer to the decision boundary $f = 0$. The number of samples in the adaptive training set is represented by $N_a$, and $\sigma(x)$ is the sigmoid function.

### 4.6 Second Pass Learning

The second pass aims to learn the SDF to the C-space boundary. Additional to the first pass, we maintain a signed distance training batch $(\mathbf{Q}_{\text{sd}}, D)$, which keeps growing as the training moves on. At each iteration, we uniformly generate configurations $\mathbf{Q}_{\text{sd}_i}$ for signed distance samples, and query the model for signs and the input point cloud for their signed distances $D_i$. Then we use the accumulated signed distance batch to help guide the neural network to learn the SDF to the boundary.

**Signed Distance Query:** For each configuration $\mathbf{Q}_{\text{sd}_i}^{(k)}$ from the signed distance samples, we approximate the closest distances $D_i^{(k)}$ by finding the closest samples in the point cloud of the opposite sign,

$$D_i^{(k)} = \begin{cases} \min_{\mathbf{q} \in \mathbf{Q}_{\text{neg}}} \left\| \mathbf{q} - \mathbf{Q}_{\text{sd}_i}^{(k)} \right\|_2 & \text{if} \quad t(\mathbf{Q}_{\text{sd}_i}^{(k)}) = 1, \\ -\min_{\mathbf{q} \in \mathbf{Q}_{\text{pos}}} \left\| \mathbf{q} - \mathbf{Q}_{\text{sd}_i}^{(k)} \right\|_2 & \text{if} \quad t(\mathbf{Q}_{\text{sd}_i}^{(k)}) = -1. \end{cases} \tag{10}$$

**Second Pass Loss Function:** The loss $L$ of the second pass is composed of three terms: sign loss, eikonal loss and signed distance loss, i.e.,

$$L = L_{\text{sign}} + \lambda_1 L_{\text{eik}} + \lambda_2 L_{\text{sd}} . \tag{11}$$

The sign loss and eikonal loss are the same as in the first pass. The signed distance loss $L_{\text{sd}}$ penalizes the difference between the predicted distances and the reference distances, and it takes the accumulated signed distance batch $(\mathbf{Q}_{\text{sd}}, D)$ as input. Since the signed distances obtained from the point cloud are only approximations, we apply a weight that measures the confidence to each signed distance sample. Suppose there are $N_{\text{sd}}$ samples in the signed distance batch, then the weighted signed distance loss becomes

$$L_{\text{sd}} = \sum_{k=1}^{N_{\text{sd}}} S(w(D))_k \left( f(\mathbf{Q}_{\text{sd}}^{(k)}) - D^{(k)} \right)^2 , \tag{12}$$

$$\text{where} \quad S(x)_i = \frac{e^{-x^{(i)}}}{\sum_{j=1}^{N_{sd}} e^{-x^{(j)}}} \ . \tag{13}$$

The function $w(x)$ gives a trusting weight that measures the confidence according to the input signed distance.

**Trusting Weight:** Since the point cloud is an approximated and discretized representation of $T_{\text{bound}}$, the distance computed from the point cloud is an approximation of the ground truth signed distance. Thus, we assign a trusting weight $w(x)$ with each signed distance sample. The trusting weight is set based on the intuition that when the queried configuration is far from the collision boundary and the approximated distance is large compared to the granularity of the point cloud representation, the error caused by the distance approximation can be ignored. In this sense, we map the signed distance to a piecewise weight function

$$w(x) = \begin{cases} 1 & x \leq -\eta_2 \ , \\ \frac{1}{2}\left(\cos\frac{x+\eta_2}{\eta_2-\eta_1}\pi+1\right) & -\eta_2 < x \leq -\eta_1 \ , \\ 0 & -\eta_1 < x < \eta_2 \ , \\ \frac{1}{2}\left(\cos\frac{x-\eta_2}{\eta_2-\eta_1}\pi+1\right) & \eta_1 \leq x < \eta_2 \ , \\ 1 & x \geq \eta_2 \ . \end{cases} \tag{14}$$

Note that $\eta_1$ serves as the distance threshold where the learner starts to trust, and $\eta_2$ is the threshold of the distance getting fully trusted. Given our sampling domain is a $2^r$ hyper cube in the range of $[-1,1]$ at each dimension, $\eta_1$ and $\eta_2$ are set as

$$\eta_1 = \frac{2}{(N_{\text{pos}}+N_{\text{neg}})^{\frac{1}{r}}} \ , \tag{15}$$

$$\eta_2 = \alpha\eta_1 \ , \tag{16}$$

where the $N_{\text{pos}}$ and $N_{\text{neg}}$ are the number of samples in positive point cloud and negative point cloud, and $\alpha$ is a user defined hyperparameter (we set $\alpha = 10$ in our tests). This is a function symmetric with respect to $x = 0$ given that the weight is purely based on the unsigned distance. The weight function produces weight 0 when the unsigned distance is smaller than $\eta_1$ and it produces weight 1 when the unsigned distance is larger that $\eta_2$, indicating we fully trust the provided distance, and a cosine function interpolates the weights in between.

## 5 REAL-TIME SIMULATION

Our contribution in real-time simulation consists of real-time SCD and collision response. For self-collision detection, the trained neural network $f(\mathbf{q})$ is used to replace the algorithmic methods, and we evaluate the SDF function instead of observing the geometry of the model. The collision response includes collision handling between pairs of reduced models and the self-collision response of each model where the network gradient $\nabla f(\mathbf{q})$ is used. The collision response forces are generated by first forming constraint Jacobian matrices that define the contact constraints and then solving for the Lagrange multipliers that represent the response forces.

### 5.1 Real-time Self-collision Detection

During the real-time simulation, we need to detect whether the model is in self-collision at each time step. Instead of resorting to traditional geometrical intersection tests, we evaluate the learned SDF function $f(\mathbf{q})$. Although the prediction boundary $f(\mathbf{q}) = 0$ does not completely align with the ground truth, it can still work well for self-collision detection because the slight misalignment of the boundary is not easily visible in the form of geometrical self-intersection.

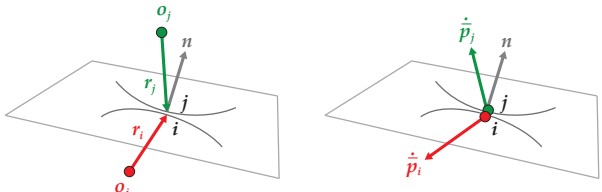

Figure 4: Diagram showing contact between objects $i$ and $j$, with contact point positions show at left, and velocities at the contact point shown at right.

In each time step, we plug the current deformation configuration $\mathbf{q}$ into the evaluation function. If $f(\mathbf{q}) > 0$, the model is considered self-collision free, regardless of the actual shape of the model. If $f(\mathbf{q}) < 0$, the model is considered in self-collision, then we need to compute the configuration velocity update caused by the self-contact constraint, which will be discussed in the next section.

### 5.2 Real-time Collision Response

Our real-time collision response is based on the contact constraint used by Erleben [7] for solving rigid body contacts. Since we simulate reduced models in a way that mixes rigid body motion and elastic deformation, we can easily extend the rigid body contact to reduced elastic body contact by adding extra entries to the contact constraint matrix to thereby incorporate deformation.

We include the gradient of the learned function into the constraint matrix to form self-collision contact constraints. We can thus solve for the configuration velocity update such that in the next time step the configuration velocity is not taking the configuration deeper into the collision space.

#### 5.2.1 Mix of Rigid and Elastic Motion

Our simulation of a reduced model consists of the rigid motion (translation and rotation) of the center of mass (COM) and the reduced elastic deformation of the model. Following Barbic et al. [4], we make the simplification of not coupling rigid and deformable motion, although this would not be difficult to implement [19]. The origin of the COM frame is set to be the center of mass of the model at the rest shape. For each vertex, we compute the deformed position in the COM frame and then transform it into the world frame to get the world position

$$\mathbf{x}_{\text{w}} = \mathbf{R}(\mathbf{x}_0 + \mathbf{U}\mathbf{q}) + \mathbf{p} \ , \tag{17}$$

where $\mathbf{R}$ and $\mathbf{p}$ are the rotation matrix and world position of the center of mass, and $\mathbf{x}_0$ is the initial position of the vertex in COM frame. The rotation matrix $\mathbf{R}$ is a matrix form of the axis-angle rotation representation $\theta \in \mathbb{R}^3$, and is obtained by Rodrigues' formula.

In the following discussion, we use a generalized coordinate $\tilde{\mathbf{x}}$ to represent the rigid motion and reduced deformation of the model,

$$\tilde{\mathbf{x}} = \begin{bmatrix} \mathbf{p} \\ \theta \\ \mathbf{q} \end{bmatrix} \in \mathbb{R}^{6+r} \ . \tag{18}$$

The approximation made here is that we disregard the rotational inertia change due to deformations. Since our focus is on collision detection and response, we make the approximation to enable a simple extension from rigid body contact constraint to reduced model contact constraint. In practice, the reduced model still behaves naturally after applying the approximation in our simulation.

### 5.2.2 Reduced Model Contact Constraints

In order to solve the contact between two objects, the relative velocity of the contact point should be zero or cause separation in the normal direction, and this inequality is expressed in the form of a row of the contact constraint matrix. Figure 4 shows an example contact, where $\mathbf{n}$ is the normal of the tangent plane pointing from $i$ to $j$, and $\mathbf{r}_i$, $\mathbf{r}_j$ are relative positions of the contact point to COMs of the two objects. We can write velocity level constraint into

$$\underbrace{\begin{bmatrix} -\mathbf{n} & -\mathbf{r}_i \times \mathbf{n} & -\mathbf{n}\mathbf{R}_i\mathbf{U}_i & \mathbf{n} & \mathbf{r}_j \times \mathbf{n} & \mathbf{n}\mathbf{R}_j\mathbf{U}_j \end{bmatrix}}_{\mathbf{J_k}} \underbrace{\begin{bmatrix} \mathbf{v}_i \\ \boldsymbol{\omega}_i \\ \mathbf{u}_i \\ \mathbf{v}_j \\ \boldsymbol{\omega}_j \\ \mathbf{u}_j \end{bmatrix}}_{\dot{\mathbf{x}}} \geq \mathbf{0} , \quad (19)$$

where $\mathbf{v}_i$ and $\mathbf{v}_j$ are the translation velocities of center of mass, $\boldsymbol{\omega}_i$ and $\boldsymbol{\omega}_j$ are the angular velocities. Additionally, $\mathbf{U}_i$ and $\mathbf{U}_j$ are the interpolated deformation bases of the points at the contact point of the models, $\mathbf{u}_i$ and $\mathbf{u}_j$ are the velocities of the deformation configurations, which represent how fast the model is deforming.

### 5.2.3 Reduced Self-collision Constraints

When the configuration of a reduced model has a negative signed distance, the model is determined to be in self-collision. This can be considered as a violation of the self-collision constraint $f(\mathbf{q}) \geq 0$.

In order to move the configuration to a contact free area in C-space, we can make use of the SDF gradient $\nabla f(\mathbf{q})$ which generally provides the direction to the closest point on the self-collision boundary. The goal of the self-collision handling when $f(\mathbf{q}) < 0$ is to finally take the deformation configuration into a collision free area in C-space, so the signed distance evaluation in the next time step should be no smaller than the current one

$$f(\mathbf{q} + \mathbf{u}\Delta t) \geq f(\mathbf{q}) , \quad (20)$$

where $\mathbf{u}$ denotes the configuration velocity and $\Delta t$ is the time step size in the simulation. Expanding left-hand side using a first order Taylor series gives us

$$f(\mathbf{q}) + \nabla f(\mathbf{q})^T \mathbf{u} \Delta t \geq f(\mathbf{q}) , \quad (21)$$

$$\nabla f(\mathbf{q})^T \mathbf{u} \geq 0 , \quad (22)$$

which defines the self-collision constraint in velocity level. Then we can add an additional row in the constraint Jacobian matrix, and put the SDF gradient in the block corresponding to the model in the whole system:

$$\underbrace{\begin{bmatrix} \mathbf{0}^T & \mathbf{0}^T & \nabla f(\mathbf{q})^T \end{bmatrix}}_{\mathbf{J_k}} \underbrace{\begin{bmatrix} \mathbf{v_i} \\ \boldsymbol{\omega_i} \\ \mathbf{u_i} \end{bmatrix}}_{\dot{\mathbf{x}}} \geq 0 . \quad (23)$$

The self-collision response generated by this constraint matrix using the SDF gradient takes the configuration to the self-collision boundary in approximated shortest C-space distance. This may not be the fastest way to bring the model out of self-intersection considering the extremal points in intersection, because the Euclidean distance in C-space does not correspond to the distance of the extremal points in a self-intersection. However, the self-collision response using this method is plausible during simulation.

## 6 RESULTS

We perform multiple tests on different models with our two-pass active learning algorithm to show the performance of the learned SDF neural network function. First, we perform a first-pass algorithm on different sizes of the neural networks to have a general knowledge of the C-space complexity of different models. Then, we test the performance of our trained SDF, including quantified scores of the performance and visualizing some of the trained SDF and its ground truth. Finally, we discuss some animation results when applying the SDF function to real-time self-collision detection and handling.

### 6.1 Network and Boundary Complexity

We perform grid tests on the expressiveness of the neural network sizes, showing the complexity of the model's collision boundary, so that we can properly choose the sizes of the neural networks. In this set of tests, we only perform the sign accuracy tests on the neural networks that are trained in the first pass learner. This is because in this test we do not need the signed distance and its gradient, and what we need is just the sign accuracy test of the trained network to see how well it fits the collision boundary. Performing learning of the first pass is enough to fit the neural network to the collision boundary and see its expressiveness.

The experiments are conducted on each model we plan to learn, and the tests span the number of modes from 3 to 7. The network structures consist of 1 to 3 hidden fully-connected layers, and each of the hidden layers has the same layer size which spans from 10 to 100. The input layer is of the same size as the model's deformation mode number and takes configuration $\mathbf{q}$ as input. The activation functions for all the hidden layers are ReLU functions because the ReLU activation provides fast learning as it reduces the likelihood of gradient vanishing and is used commonly in deep learning.

We spend 500 iterations uniformly generating $\mathbf{Q}_{\text{pool}_i}$ for exploitation. In some cases where the network training becomes stuck in a local optimum and gives extremely low sign prediction accuracy, we do multiple tests and report the best test accuracy. The sign prediction accuracy is plotted in Figure 5. We can observe that for the exact same network architecture, the test sign accuracy decreases when the model has more modes, which indicates that the expressiveness of the neural network is less likely to be capable of representing the collision boundary. This suggests that the C-space boundary becomes more complex when there are more modes, and in turn requires a more complicated neural network to represent. However, the increase in the number of modes results in more accuracy decrease for the snake than for the bunny. This is probably because the addition of new modes for the snake model enables new collision between geometry parts which cannot deform to contact with the old deformation basis. On the other hand, the new modes of the bunny model are just wiggles of the geometry, which does not complicate the self-collision boundary too much except for adding a new dimension.

Another observation is that the C-space boundary can be represented by a simple neural network. For both of the models, increasing the hidden layer number from 2 to 3 while keeping the layer size fixed only slightly improves the sign accuracy, which leads us to believe that 2 hidden layers is sufficient for the bunny, the snake, and other similar models. For a simpler model like the bunny, we can get at least 95% sign prediction accuracy in approximating its 7D collision boundary, using a simple neural network with 2 hidden layers, and having 50 or more nodes in each of the hidden layers. For the snake model with 7 degrees of freedom, the sign accuracy reaches around 93% with a simple network with 2 hidden layers, having 70 or more nodes in each of the hidden layers. In order to select the best layer size to set up the neural network when the change in layer size does not significantly affect the accuracy, we tend to pick the point at the knee of the graph. In terms of learning the models with 7 modes, the architecture picked for the bunny and

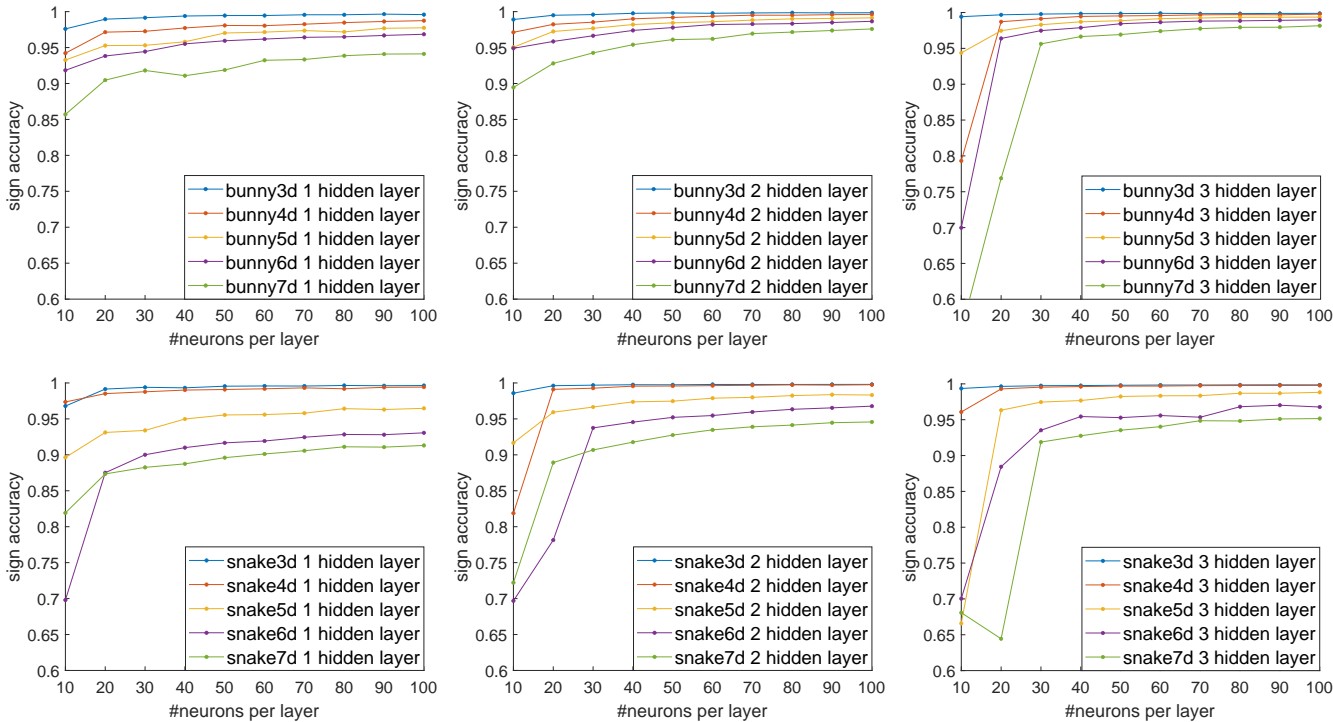

Figure 5: Evaluations of necessary network complexity for reduced deformation dimension varying between 3 and 7. The bunny model (top) is generally easier to learn than the snake model (bottom), and accuracy for high dimensional reduced spaces can be higher with additional hidden layers (second and third column).

the snake appears to be the same, which is a fully connected network with 2 hidden layers and 70 nodes.

## 6.2 SDF Quality Measurement

We also show the performance results of the neural networks trained by our two-pass algorithm. This includes two tests: visualized learning results of 2D SDFs and quantified performance scores of the learned C-space SDF of reduced models.

We test our approach on multiple models, and train our neural network to learn the target SDF. For both passes of each model, we spend 500 iterations in the training. The network used in both passes has 2 hidden layers of the same size 70, with ReLU activation function used. The same network architecture is used in this test so that we can compare the results between different models or the same model with a different number of modes. During each iteration, we train 10 epochs with the initial learning rate set to 0.001.

### 6.2.1 2D SDF

We first present the test results of learning 2D SDFs. In this test, the self-collision boundary is defined by binary images. We visualize the trained SDF and compare it with the ground truth SDF that we compute exactly by finding the closest point on the boundary. Note that in training we still only use the sign labels of samples, and the ground truth distance is only used in visualization.

We train and visualize the SDF defined by an Apple logo where the decision boundary is generally smooth and a Twitter logo where a lot of sharp features exist and the decision boundary is generally harder to learn. The test result is visualized in Figure 6. By qualitative comparisons between the trained SDFs and the target SDFs, we can see the two-pass algorithm works very well for 2D examples. The trained neural network not only provides a very good approximation of the boundary, capturing the sharp features, but also has smooth SDF gradient over the domain despite some differences

compared to the target SDF. The success in these examples suggests that our learning approach does have the ability to detect and learn the sharp features in C-space boundary, and it is also feasible to directly learn the SDF instead of just the boundary representation. This success in 2D examples also gives us the encouragement to extend our tests to higher dimensions.

### 6.2.2 Reduced Model C-space

In order to get target signed distances for quality measurement of the predicted signed distance, we spend 5000 iterations on the first pass to generate a denser point cloud. Then, the dense point cloud is used in computing the target signed distance value. Note that this can be accurate when the model has a small number of deformation bases but less accurate as the deformation degrees of freedom increases.

In testing the trained neural network, we uniformly generate $N_{\text{test}} = 50000$ samples, and compute the error $e_{\text{sd}}$ between the predicted signed distance and queried signed distance on the dense point cloud. We also compute the sign prediction accuracy $\eta$, gradient size error $e_{\text{grad}}$ and gradient size standard variance $\sigma_{\text{grad}}$,

$$e_{\text{grad}} = \frac{1}{N_{\text{test}}} \sum_{k=1}^{N_{\text{test}}} \left| \left\| \nabla f(Q_{\text{test}}^{(k)}) \right\|_2 - 1 \right|, \qquad (24)$$

$$\sigma_{\text{grad}} = \sqrt{\frac{1}{N_{\text{test}}} \sum_{k=1}^{N_{\text{test}}} \left( \left\| \nabla f(Q_{\text{test}}^{(k)}) \right\|_2 - \mu_{\text{grad}} \right)^2}, \qquad (25)$$

$$\text{where} \quad \mu_{\text{grad}} = \frac{1}{N_{\text{test}}} \sum_{k=1}^{N_{\text{test}}} \left\| \nabla f(Q_{\text{test}}^{(k)}) \right\|_2. \qquad (26)$$

The sign prediction accuracy $\eta$ is used to measure the ability of the trained neural network to detect self-collision of the reduced model. Gradient size error $e_{grad}$ and gradient size standard variance

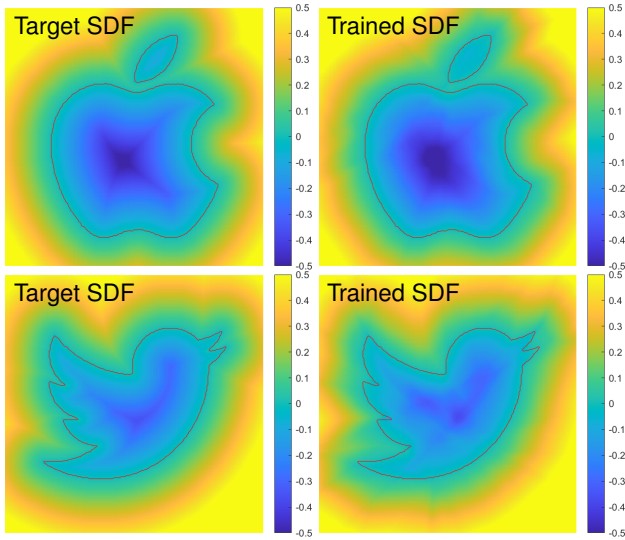

Figure 6: Visualization of two dimensional SDF learning tests show excellent accuracy at the boundary, and less accurate distances in the interior.

Table 1: Learned C-space SDF function quality measurement, with $\eta$ measures the sign predictions accuracy, $e_{sd}$ measures the signed distances error, $e_{grad}$ and $\sigma_{grad}$ measure the gradient error and variance computed from 50000 samples.

| model name | # modes | $\eta$ (%) | $e_{sd}$ | $e_{grad}$ | $\sigma_{grad}$ |
|---|---|---|---|---|---|
| snake | 3 | 99.77 | 0.0139 | 0.0866 | 0.1192 |
| | 4 | 99.62 | 0.0206 | 0.1039 | 0.1343 |
| | 5 | 98.51 | 0.0439 | 0.1995 | 0.2594 |
| | 6 | 96.11 | 0.0834 | 0.2140 | 0.2728 |
| | 7 | 95.20 | 0.1151 | 0.2542 | 0.3226 |
| bunny | 3 | 99.93 | 0.0122 | 0.0840 | 0.1170 |
| | 4 | 99.50 | 0.0312 | 0.1911 | 0.2505 |
| | 5 | 98.69 | 0.0436 | 0.2305 | 0.2949 |
| | 6 | 96.95 | 0.0746 | 0.2583 | 0.3249 |
| | 7 | 94.87 | 0.1036 | 0.3205 | 0.3960 |
| bracelet | 3 | 99.77 | 0.0189 | 0.1191 | 0.1595 |
| | 4 | 99.15 | 0.0344 | 0.1754 | 0.2263 |
| | 5 | 98.12 | 0.0507 | 0.1886 | 0.2403 |
| | 6 | 96.39 | 0.0738 | 0.2025 | 0.2538 |
| | 7 | 94.68 | 0.1099 | 0.2263 | 0.2805 |

$\sigma_{grad}$ denote how well the learned SDF is representing the Euclidean distance in the C-space. They are included to rule out the cases where the decision boundary fits well but the gradient is dramatically changing in the C-space. Smaller values of $\sigma_{grad}$ and $e_{grad}$ mean that the gradient is smooth in the learned SDF and thus can potentially give good directions when queried to solve self-collision.

We test our two-pass learning algorithm on bunny, snake and bracelet model. The bracelet and bunny have relatively simpler C-space boundaries. For the bracelet model, it can only generate collisions between the two ends of the crack. For the bunny model, the collision only happens between the two ears and the back. However the snake model has a more complex boundary and SDF as well because it has many adjacent coils that can have collisions.

The test results are shown in Table 1. The test sign accuracy for models with 3 or 4 modes can reach more than 99%. The sign accuracy goes down as we increase the number of modes, and the sign accuracy becomes around 95% at 7 modes. This is not so ideal considering the test samples are uniformly sampled in C-

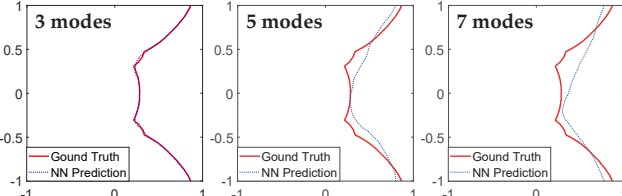

Figure 7: Visualization of a 2D slice (first two modes) of higher dimensional C-space boundaries for the bracelet model, with all other coordinates set to zero. The NN prediction struggles to fit the boundary in higher dimensional spaces.

space, so a lot of them are far from the ground truth boundary, where it is easy to correctly predict the sign. Figure 7 shows the visualization of the 2D decision boundary of trained neural networks of the bracelet model, providing some intuition into the sign accuracy in Table 1. For a bracelet with 3 deformation modes, the sign accuracy 99.77% indicates a very well-aligned collision boundary between the predicted one and ground truth. When the number of modes increases to 7, the 94.68% accuracy prediction boundary is less ideal, and becomes a coarse approximation of the ground truth.

The signed distance error for models with 3 or 4 modes is around 0.01, which is good considering we are testing in the range of a $[-1, 1]$ hyper-cube. It becomes approximately 0.1 when the number of modes reaches 7.

### 6.3 Summary

In our experiments, we perform sign accuracy tests on neural networks with different architecture and different sizes. Through this test, we can have an intuition of the self-collision boundary complexity of different models, and we can also reasonably choose the number of hidden layers as well as the size of the hidden layers. We also measure the quality of the trained SDF which is learned by applying the two-pass learning method, and apply the learned SDF in real-time reduced model simulation for self-collision detection and response.

Our method works very well in learning the SDF in low dimensional configuration space. The 2D examples show that the two-pass learning algorithm not only successfully learns the representation of the boundary, but also provides smooth SDF within the 2D configuration space. The simulation examples of the spring and bracelet with 3 modes also show that the trained neural networks provide good self-collision approximation and generate reasonable self-collision response. However, when the target SDF has more dimensions, our learning method has a difficult time to learn a good approximation of the boundary as well as the SDF. This can be seen in supplementary video for the 7 mode snake and the 10 mode bunny.

### 7 CONCLUSION

In this work, we propose the concept of self-collision boundary and C-space SDF. We also propose and implement a two-pass active learning algorithm that approximates the C-space SDF with a neural network that is trained on samples with only sign labels. The main idea is to use exploration and exploitation criteria to pick the most informative samples so that the convergence speed is improved. We also use an eikonal loss term and approximated signed distances to ensure that the neural network is not only skilled at determining the boundary, but also representing the distance to the boundary. Moreover, we propose a method to make use of the trained SDF function in self-collision detection and setting up a self-contact constraint matrix with the gradient.

## 7.1 Advantages and Limitations

Our learning approach uses active learning to select samples for training, which helps improve sample efficiency and reduce the number of SCD queries. We have shown that our method can do a great job at learning the collision boundary of models with a small number of modes, and can reconstruct the signed distance field very well in 2 dimensional space. The learning is purely based on samples with only sign labels, which helps us bypass the dilemma where we have neither the ground truth distances nor a stable way to get approximated signed distances. Furthermore, the cost of evaluating the learned SDF to detect self-collision is constant, and making use of the gradient for self-contact handling is compatible with standard constraint solving methods.

Our work also has important limitations. One limitation is that currently our learning method only works well when the model has a small number of modes. Although we reduce the adverse impact from dimension increase by selecting informative samples, the curse of dimension still persists and makes it hard to learn the SDF in high dimensional space. For simple models that require a small number of modes to deform, our algorithm works nicely and produces great results. But the learner still struggles to learn the collision boundary in high dimensional space, which is the case when the model needs a large number of modes to produce plausible deformation. Another limitation is that our way of moving out of self-contact along the SDF gradient cannot take into account frictional contact. Although this method generates plausible self-collision solutions, the relative velocity between intersecting parts is not necessarily along the normal direction of the contact plane. This means by only evaluating the signed distances in the configuration space, our method cannot provide the information of the contact normals, and thus we can not set up constraints for frictions caused by self-collision.

## 7.2 Future Work

There are many possible ways to overcome existing limitations. One possible improvement is looking for a new sampling strategy to improve the reliability and accuracy in learning high dimensional subspace. Currently the exploitation samples are selected within a data pool $\mathbf{Q}_{\text{pool}_i}$ that is generated at each iteration. The picked exploitation samples are close to $f(\mathbf{q})$ only if the data pool has samples close to the boundary, which cannot be easily achieved when the C-space dimension is high. Instead, we can look for a method that can generate $\mathbf{Q}_{\text{pool}_i}$ whose samples are mostly close to the boundary. One of the ideas is making use of the network gradient, and using Newton's method to find roots for $f(\mathbf{q}) = 0$.

We can also further extend our work to other applications. We can consider inter-object collision detection as a self-collision problem by taking all the objects as a whole, and learn the C-space. Given that our learning method is good at reconstructing 3D space, we can also possibly make use of our learning method in mesh reconstruction.

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
