# OpenReview forum: "Active Learning Neural C-space Signed Distance Fields for Reduced Deformable Self-Collision"
_graphicsinterface.org/Graphics_Interface/2022/Conference — GI 2022_

### Official Review · Reviewer_8Ge2 · 2022-04-08
**An interesting and novel method for collision computation**

**Rating:** 5
**Confidence:** 3

**Review:**

This paper proposes a two-pass active learning method to learn the neural C-space SDF for self-collision detection and deformable collision response. Therefore, the complex collision detection or collision response simulation can be turned into a problem to evaluate the signed distance or gradients in the learned low-dimensional C-space. Overall, I think the presented method is very interesting and novel for collision computation.

Pros:
1) The proposed two-pass active learning method can learn the SDF with informative gradients from the estimated signs of the sampled configurations.
2) The learned C-space SDF is useful in the collision simulation applications, including real-time self-collision detection and response.

Cons:
1) My main concern is about the number of modes. The proposed method achieves good accuracy when the mode number is smaller than 7. I'm not sure whether it's enough for the common deformable models. I'd like to know more about the computation of the modes, such as the LMA, modal derivatives and manual selections mentioned in Section 3.

2) There are some writing errors and unclear descriptions to be improved. For example, "so the it's samples" in Section 4.5 ("Point Cloud Collection" paragraph), "Figure??" in Section 3. And what does it mean by "During each iteration, we train 10 epochs with the learning rate set to 0.001" in Section 6.2?

3) As for the evaluation, except for the metrics in Table 1, it's better to compute the direction error of the predicted gradients.

---

### Official Review · Reviewer_1U1D · 2022-04-12
**Interesting approch, which could be evaluated in more detail**

**Rating:** 5
**Confidence:** 3

**Review:**

# Summary and contributions

The paper introduces a semi-supervised method to learn the signed distance function (SDF) of the configuration space (C-space) of a reduced model. In this context, the SDF represents the distance of a reduced configuration to a boundary separating deformations where the model is self-intersecting. The SDF is learned with a neural network. The training of the neural network is divided into two passes. In the first pass, a set of point samples with their signs is generated automatically with self-collision tests and used for training the network incrementally. In the second pass, the point samples and newly-extracted SDF samples are used for training the network and thus learning the SDF. The learned SDF is used for real-time self-collision detection and response, where the SDF approximates the configuration space.

In terms of contributions, the paper follows the idea of Pan et al. [17] of using active learning for learning a representation of the C-space of rigid bodies. Some of the additions to Pan et al. include the use of a neural network with Eikonal terms to constrain the learned gradient [2], and the inclusion of elastic motion in the model, captured by the reduced configuration, in addition to rigid motion.


# Evaluation of the paper

Strengths:
- The paper introduces a learning-based pipeline for efficiently checking self-intersections in deformed models.

- The method can also be used for collision resolution to some extent.

- In comparison to Pan et al. [17], the method uses state-of-the-art machine learning (deep neural networks), and adds elastic motion to the rigid motion.

Weaknesses:
- The main weakness of the paper is that evaluation could have been more detailed:

- The paper could have included a comparison to traditional geometric approaches in terms of execution time and memory used, to show the advantages of the method more clearly.

- The method could have performed some evaluation on a simulated collision test. Table 1 shows that the method learns a model with 3 modes more accurately than with a higher number of modes. However, it would be expected that a model with a higher number of modes would be more accurate when used in practice. Thus, besides evaluating the test accuracy on samples for a given number of modes, an experiment could also evaluate the global accuracy across modes for some test cases.

- The paper could include ablation experiments, especially an experiment training the network with and without the gradient (Eikonal) terms to justify their use.

In summary, the paper is an initial exploration into the proposed direction, but could benefit from more conclusive evaluation. In addition, the contribution is a bit small as it is the adaptation of previous work to the proposed problem.

## Other comments:
- Section 5.2.3 assumes that the closest contact-free configuration in the reduced model corresponds to the closest contact-free configuration in 3D space. Are there guarantees that this is a reasonable assumption?


# Clarity of presentation

The paper is in general well-written and clear.

Some comments on the text:

- The introduction could include more citations, e.g., a citation to model reduction in animation, the BD-Tree, and so on for other concepts introduced.

- The introduction could also clarify the contributions of the work more clearly, explaining the differences of the work to Pan et al. [17].

- W(x) in Eq. (9) looks like a softmax. Same for Eq. 13.

- I understand that the paper claims to use “active learning” as this term was also used in Pan et al. [17]. However, to me this appears to be more of a self-learning method.

## Typos that I encountered while reading:

- The SALD reference is called “SAL++”.

- A figure reference is missing on page 2: (Figure ??).

- “so the it’s samples” -> “so its samples”

- “Then, the dense point cloud in computing the target signed distance value” -> “is used”?

- Caption of Table 1: e_grad and sigma_grad measure the signed distances error -> “gradient error”?

---

### Official Review · Reviewer_jHPk · 2022-04-13
**Review of neural Cspace SDF**

**Rating:** 4
**Confidence:** 2

**Review:**

This paper proposes an active learning method to predict the signed distance function (SDF) of a shape given the parameters in the reduced space. This approach can be used to detect self-collision in real-time for reduced model deformation.

I am leaning towards rejecting this submission. I find it difficult to assess the magnitude of the contributions. I recommend adding more evaluations on (1) the efficiency of using active learning and (2) comparing the performance of neural C-space SDF with baselines (such as using BVH).

In terms of my first concern about active learning, I agree with the claim that active learning can query more informative samples so theoretically it could learn a better SDF compared to other sampling methods given the same amount of points. Thus active learning is extremely powerful if getting ground truth data is extremely expensive. However, in the case of learning SDF, it is unclear to me whether querying signed distance from a mesh (with spatial hierarchy acceleration) is a process that is expensive enough to utilize active learning. I would recommend including a comparison between learning SDF with active learning vs learning SDF with heuristic sampling (such as section A.2 in "Neural Geometric Level of Detail- Real-time Rendering with Implicit 3D Shapes"). If the active learning approach can learn a comparable SDF with less wall-clock time, then this contribution would be more compelling.

In terms of my second concern about the performance. I am also curious about how much speed up the neural C-space SDF gives us, compared to traditional approaches such as using BVH. As pointed out in the paper, a neural approach usually suffers from accuracy and generalizability (e.g. a bigger reduced space), it would be more convincing to know the magnitude of the speed up when using neural approaches.

I also noticed that there is a missing figure reference in section 3 (Figure ??)

---

### Official Review · Reviewer_atQy · 2022-04-19
**Interesting and novel work combining reduced deformable models with neural representations**

**Rating:** 8
**Confidence:** 3

**Review:**

This paper presents an approach to self-collision using configuration space SDFs represented as neural networks. The networks can detect intersecting states, return signed distances, and use backpropagation to compute gradients that can be treated as contact normals in a traditional simulation pipeline.

The paper is interesting, and contains a few novel pieces that extend previous work, by combining SFDs with a reduced model, and around the active-learning approach. The paper is generally well written, but there were a few areas that I thought could use some clarification:

"Nerual" in Figure 1.

In the introduction it was not totally clear what the  "C-space SDF" is, it might be helpful to give a more precise definition earlier on so it's easier to follow the rest of the discourse, e.g.: traditional SDFs map a point in Euclidean space to the distance to an object's boundary. C-Space SDF's map from configuration space to what? The distance between the object and itself? Or a mapping from configuration space to just the sign representing intersection / non-intersection? Possibly warrants a formal function definition to make this clear.

Section 3. broken Figure ?? ref.

Section 4.1 "with great success."

Do SDFs still have unit magnitude in these non-Euclidean spaces? I had assumed that if you had an SDF function in Euclidean space (with unit gradient) then when you transformed to the C-Space via. the U matrix then the norm will also change?

Why not use the exact distance instead of distance to closest sample with opposite sign? e.g.: SDF for a deformed mesh can be computed quite quickly using a BVH / fast winding numbers. Seems this would avoid the need for the Trusting Weight heuristic?

Is there a reference for this kind of floating frame model used in 5.2.1? (I recall Shabana's text books have some discussion of this approach).

Eq. 19, why using the -> vector notation here? I think it's inconsistent and unnecessary?

Section 6.1, "we do multiple tests and report the best test accuracy", presumably changing random seed each time?

Overall I think this paper is exploring an interesting area (combining reduced order models with neural represetnations) that seems quite likely to become practical, and I am fairly sure this will inspire some follow up work. I am happy to recommend acceptance, but more details on training would help reproducibility, e.g.: optimizer used (Adam, etc), learning rate, etc. Also some performance numbers that break down the cost of forward inference, and backward gradient calculation at runtime would be useful.

---

### Decision · Program_Chairs · 2022-04-19

Accept